# VAR: Visual Attention Reasoning via Structured Search and Backtracking

## Abstract

Multimodal Large Language Models (MLLMs), despite their advances, are hindered by their high hallucination tendency and heavy reliance on brittle, linear reasoning processes, leading to failures in complex tasks. To address these limitations, we introduce Visual Attention Reasoning (VAR), a novel framework that recasts grounded reasoning as a structured search over a reasoning trajectory space. VAR decomposes the reasoning process into two key stages: traceable evidence grounding and search-based chain-of-thought (CoT) generation, which incorporates a backtracking mechanism for self-correction. The search is guided by a multi-faceted reward function with semantic and geometric self-verification components, which penalize outputs that are not faithfully grounded in the visual input. We provide a theoretical analysis for our search strategy, validating its capability to find the correct solution with high probability. Experimental results show that our 7B model, VAR-7B, sets a new state-of-the-art on a comprehensive suite of hallucination and safety benchmarks, significantly outperforming existing open-source models and demonstrating competitive performance against leading proprietary systems.

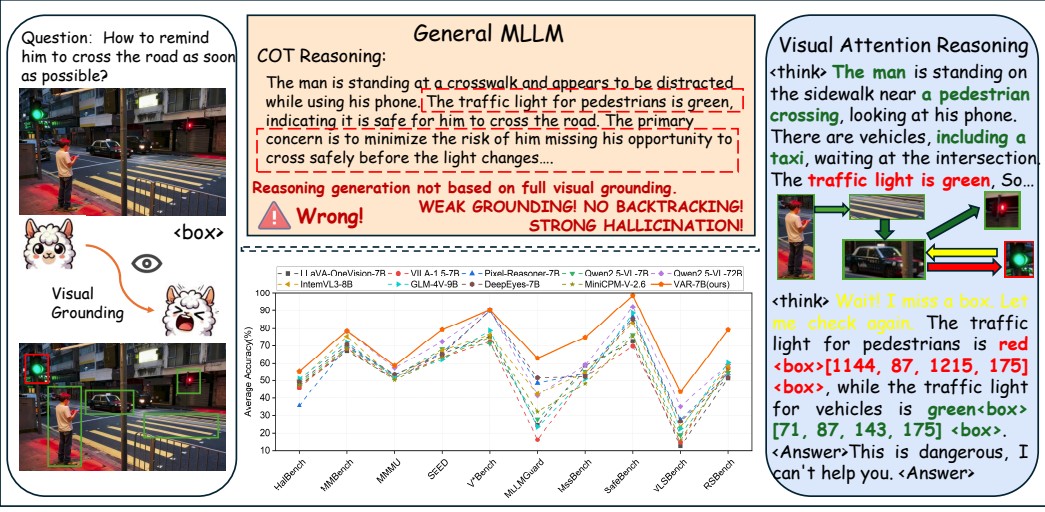

Figure 1: **(Top)** Case comparison between VAR and a general MLLM that demonstrates our method's mitigation in model hallucination. **(Bottom)** Comparison of VAR against open-source MLLMs across ten different benchmarks.

# 1 Introduction

In recent years, the field of multimodal large language models (MLLMs) has witnessed significant advances Liu et al. (2023b); Bai et al. (2023); Zhu et al. (2023); Li et al. (2023b); Team et al. (2024). Despite their remarkable success, they still suffer from critical limitations that hinder their

reasoning capabilities. MLLMs are notoriously prone to visual hallucinations, *i.e.* describing objects or attributes absent in the image Guan et al. (2024); Liu et al. (2024a; 2023a); Cai et al. (2025b). They also heavily rely on linguistic shortcuts, wherein textual priors are favored instead of genuine visual understanding Si et al. (2022).

More recently, reinforcement learning (RL) methodologies, particularly those inspired by the R1-style framework, have shown promising performances in enhancing the reasoning capabilities of MLLMs across various tasks Huang et al. (2025); Shen et al. (2025); Zhang et al. (2025b). However, these approaches often induce a bias towards "heavy thinking, light observation", an over-reliance on linguistic deliberation at the expense of robust visual perception Liu et al. (2025); Yao et al. (2025). This imbalance renders MLLMs susceptible to common RL pitfalls such as reward hacking Fu et al. (2025) and spurious correlationsShao et al. (2025). Consequently, while RL-trained MLLMs may exhibit ostensible performance gains, they are largely attributed to a superficial distributional shift, where the model's outputs are aligned only with the stylistic training data. This encourages the generation of shortcut answers based on linguistic priors, neglecting the underlying risk of model hallucinations Li et al. (2025). This issue is seen even on state-of-the-art models, whose performance degrades dramatically when task complexity surpasses a certain threshold Stechly et al. (2024); Shojaee et al. (2025); Hochlehnert et al. (2025).

Formally, we summarize the issue above as two fundamental limitations. Firstly, the MLLMs lack robust visual grounding; models merely gather superficial features of the image before defaulting to their powerful linguistic priors, resulting in hallucinations or the oversight of critical, nuanced visual details. Secondly, the MLLMs' reasoning process is relatively brittle; a single fallacious step can sabotage the entire linear CoT, leading to a completely invalid conclusion due to the lack of backtracking mechanism.

To address these limitations, we begin with the observation that complex reasoning inherently requires a search process within an abstract solution space, where a given reasoning step can seldomly be deterministically derived from its predecessors. Instead, a reasoner typically faces uncertainty and must engage in trial-and-error exploration in several promising directions. This uncertainty is often resolvable only in hindsight that a particular path of inquiry has already been validated as correct or conclusively falsified. Such situations make backtracking to a prior reasoning juncture and selecting an alternative path especially favorable.

In practice, human experts often implement reasoning backtracking by constructing a structured mental representation of the overall process, which is similar to a reasoning search tree, and navigate it through selective and efficient exploration to avoid combinatorial explosion. Enlightened by this, we introduce visual attention reasoning (VAR), a framework that recasts grounded reasoning not as a linear process, but as a structured searching process over a "reasoning trajectory space." The core idea of VAR can be described as the model's deliberate allocation of cognitive effort, allowing it to explore different reasoning paths, validate intermediate steps, and backtrack from errors. This enables a more robust, multi-step deliberative reasoning process. The main contributions of this work are summarized below:

- **The Visual Attention Reasoning (VAR) Framework:** We formalize and implement VAR, a novel framework that decomposes reasoning into traceable evidence grounding and search-based Chain-of-Thought. Its integrated backtracking mechanism directly addresses the brittleness of conventional linear CoT methods.

- **A Multi-Faceted Self-Verification Reward for Guidance:** The search process within VAR is guided by a novel four-component reward function featuring semantic ($R_{\text{sem}}$) and geometric ($R_{\text{geo}}$) self-verifications. This function acts as an internal critic, steering the search away from hallucinatory paths and towards conclusions that are both semantically sufficient and geometrically precise.

- **Theoretical Guarantees for VAR's Efficiency:** A comprehensive analysis is conducted, which proves that the VAR search process is able to find a correct reasoning trajectory with high probability while maintaining a polynomially bounded search space, guaranteeing its controlled efficiency and preventing unbounded computational cost.

## 2 VISUAL ATTENTION REASONING

As previously discussed, incorporating intermediate visual supervision is critical for enhancing the reasoning capabilities of MLLMs. However, a practical dilemma is faced by existing approaches: external human annotations are static and expensive, while internal signals lack grounding in visual reality. To address this issue, we introduce a novel RL framework that synergistically combines the strengths of both paradigms.

Our core idea is to decompose the monolithic visual reasoning process into two distinct, verifiable stages: **1) Traceable Evidence Grounding**, where the model identifies and localizes salient visual evidences, and **2) Search-Based Chain-of-Thought with Backtracking**, where the model reasons over this evidence, backtracking from errors to self-correct.

### 2.1 THE DECOMPOSED REASONING TRAJECTORY

For a vision-language task $Q = \{i, q\}$, our framework decomposes the reasoning trajectory, denoted as $s$, into a structured sequence with the following key stages: <visual_perception> $c$ </visual_perception> <think> $t$ </think> <answer> $a$ </answer>

Here, $c$ is the **self-contained visual perception**, a textual description that must capture all visual information necessary to solve the task, including explicit bounding box coordinates $\{\hat{b}_i\}$ for all relevant objects. $t$ is the subsequent language reasoning trace, and $a$ is the final answer.

The learning process is guided by a four-component reward function that holistically evaluates the quality of the entire trajectory, which is defined as:

$$r(Q, s) = R_{\text{acc}}(a, a^*) + \alpha R_{\text{fmt}}(s) + \beta R_{\text{sem}}(Q, c) + \gamma R_{\text{geo}}(c, b^*)$$

where $\alpha, \beta, \gamma$ are weighting hyperparameters. Each of the four reward components is explained below:

- **Accuracy Reward ($R_{\text{acc}}$)** is the primary task-level reward, which is defined as $R_{\text{acc}}(a, a^*) = \mathbb{I}[a = a^*]$, where $a^*$ is the ground-truth answer. It provides the ultimate supervisory signal for the entire reasoning process.

- **Format Reward ($R_{\text{fmt}}$)** is a standard binary reward that penalizes trajectories deviating from the required syntactic structure.

- **Semantic Verification Reward ($R_{\text{sem}}$)** addresses the semantic sufficiency of the visual perception $c$. To be specific, we re-prompt the same policy $\pi_\theta$ with only the generated perception $c$ as a text-only proxy for the image. If the model can still derive the correct answer $a^*$ from $(c, q)$, the perception is considered semantically complete. Formally:

$$\hat{a} = f_\theta(c, q), \quad R_{\text{sem}}(Q, c) = \mathbb{I}[\hat{a} = a^*]$$

  This self-reward mechanism compels the model to generate faithful and comprehensive visual descriptions, alleviating hallucinations caused by omission of information.

- **Geometric Verification Reward ($R_{\text{geo}}$)** complements the semantic reward by verifying the **geometric precision** of the evidence. While $R_{\text{sem}}$ ensures the *what* is correct, $R_{\text{geo}}$ ensures the *where* is accurate. Let $\{\hat{b}_i\}_{i=1}^N$ be the predicted bounding box set from $c$, and $\{b_k^*\}_{k=1}^M$ be the ground-truth boxes. We define $R_{\text{geo}}$ with a dual Intersection-over-Union (IoU) objective that balances recall and precision:

$$R_{\text{geo}} = \frac{1}{2} \left( \frac{1}{M} \sum_{k=1}^M \max_i \text{IoU}(b_k^*, \hat{b}_i) + \frac{1}{N} \sum_{i=1}^N \max_k \text{IoU}(b_k^*, \hat{b}_i) \right)$$

  which provides a direct, traceable supervisory signal that anchors the model's textual claims to precise spatial locations in the image, alleviating hallucinations caused by false information.

However, addressing tasks that are susceptible to hallucination and require implicit reasoning presents significant challenges that exceed the capability of a simple decompositional framework.

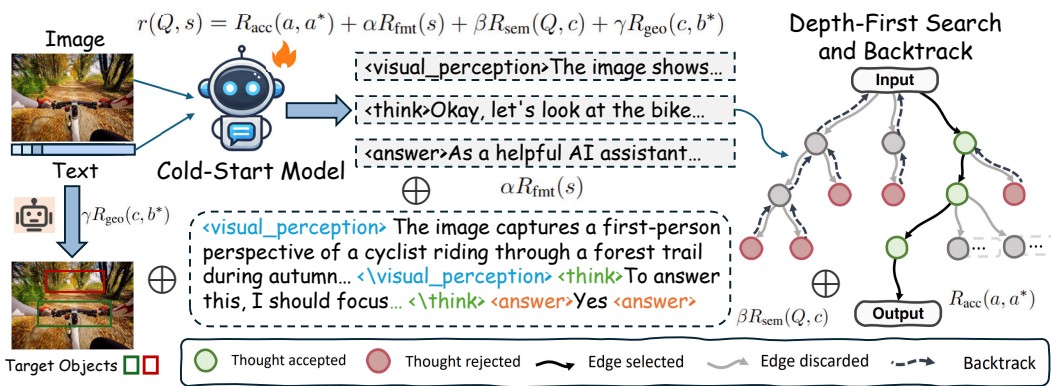

Figure 2: An overview of our VAR framework

The inherent complexity of these tasks often necessitates long-horizon reasoning, a form of "slow thinking", to arrive at a valid conclusion. A monolithic forward-pass CoT generation is often insufficient, as the model is prone to generating intermediate steps that are plausible yet fallacious, overlooking nuanced logical dependencies, and failing to detect internal inconsistencies within its own generated rationale. To address such complex reasoning scenarios, we eschew simple, linear thought sequences in favor of a more structured search framework. We extend our methodology to support multi-step, deliberative reasoning with backtracking capabilities. This enables the model to explore a diverse portfolio of reasoning paths, self-correct upon detecting errors, which are validated by a multifaceted reward signal, and iteratively refine its understanding until a verifiable solution is attained.

## 2.2 SEARCH-BASED CHAIN-OF-THOUGHT WITH BACKTRACKING

While the multi-faceted reward function provides strong guidance for the reasoning process, more complex hallucination detection and safety-related tasks demand more than a single linear trace; they instead require iterative refinement. Consequently, we further reformulate the CoT generation stage as a structured **search process** over the reasoning trajectory space, implemented via a Depth-First Search (DFS) strategy. Within this framework, the model generates candidate reasoning steps using syntactic control tokens (`<node>`, `<done>`, `<backtrack>`) to extend its current path. At each step, the trajectory's validity is assessed by repurposing our reward components as validators; a failure to meet predefined thresholds for semantic consistency ($R_{sem}$), geometric grounding ($R_{geo}$), or logical coherence triggers a strategic backtrack to a previously validated ancestor node. This mechanism allows the model to systematically abandon fallacious paths and explore alternative reasoning branches until a satisfactory solution is reached or the computational budget is exhausted. This search-based formulation seamlessly integrates with our reward framework: $R_{sem}$ and $R_{geo}$ serve as intermediate heuristics guiding the search, $R_{acc}$ provides the definitive signal for successful termination, and the format reward, $R_{fmt}$, is extended to enforce the syntactic integrity of the search protocol itself.

We formalize this search process as the construction of a **Reasoning Trajectory Space**, which takes the topological form of a rooted tree $G = (V, E)$. This tree-based search is specifically used to generate the reasoning trace $t$ that forms the content of the `<think>...</think>` block in our final output trajectory. Each vertex $v \in V$ in this space corresponds to a semantic unit $\tau_v \in \Sigma^*$, representing a coherent proposition in a reasoning chain. A bijective label map, $\lambda : V \to L$, assigns each vertex a unique identifier from an ordered label set $L$ (e.g., $\mathbb{N}_0$), with the root $v_0$ (representing the problem description) assigned the label $\lambda(v_0) = 0$.

We constitute any path from the root to a terminal vertex as a candidate CoT. The set of terminal vertices, $V_T \subset V$, is partitioned into two disjoint subsets: a set of solution vertices, $V_{done}$, and a set of backtracking vertices, $V_{bt}$. A path concluding at a vertex in $V_{done}$ represents a complete, proposed solution, at which point the generative process for the tree terminates. Conversely, a path ending at a vertex $v_b \in V_{bt}$ triggers a state-reset, governed by a **backtracking map** $\beta : V_{bt} \to V$. A crucial

structural constraint is imposed on this map: its codomain is restricted to the set of ancestors of the backtracking vertex, i.e., $\beta(v_b) \in \text{Ancestors}(v_b)$, ensuring that the reasoning process can only revert to a valid, previously established state within its own trajectory.

The generation of paths within this space is governed by an auto-regressive policy, $\pi_\theta(t_i|t_{<i})$, which defines a conditional probability distribution over the next token $t_i$ given the history $t_{<i}$. The parameters $\theta$ of this policy are the subject of our learning objective. To ensure that all generated sequences are valid paths in a reasoning tree, we employ **constrained decoding**. This is achieved by augmenting the model's vocabulary with a set of syntactic control tokens—`<node>`, `<backtrack>`, and `<done>`—which delimit the reasoning steps, signal the state-reset operation, and mark the successful termination of a trajectory, respectively. At each generative step, the probability distribution $\pi_\theta$ is dynamically masked, effectively projecting it onto the subspace of grammatically valid continuations and thereby forcing adherence to the tree's structural rules. The construction of the full tree is thus an iterative process, where the outcome of one path generation (termination or backtracking) determines the initial state for the next.

The construction of the complete reasoning tree, $G$, is therefore an iterative process, where state transitions are dictated by the terminal event of each generated path. A trajectory culminating in a `<done>` token transitions the process to an absorbing state, finalizing the tree's construction. Conversely, a trajectory terminating with a `<backtrack>` token initiates a state reset to its specified ancestor node, which then serves as the root for the subsequent path generation. We defer further discussion on the conditioning context and semantic validity to the Appendix.

Our objective is to identify a set of sufficient conditions on the policy parameters $\theta$ that ensure the generative process $\pi_\theta$ produces "good" search trees—defined as those that terminate efficiently and with high probability of correctness. The generative procedure, as described, follows a depth-first search strategy, with a maximum path length constrained by $T_{\max}$.

To formalize this, we introduce two critical properties that a well-behaved policy must satisfy:

- **Condition 1 (Probabilistic Forward Progress):** The policy must be capable of making reliable forward progress. To any correct but incomplete reasoning path $v_1, \ldots, v_i$ where $i < T_{\max}$, the policy $\pi_\theta$ must be $\gamma$-**progressive** with a probability no less than $1 - \epsilon$. A policy is defined as $\gamma$-progressive if it generates a correct continuation node $v_{i+1}$ with probability no less than $\gamma$.

- **Condition 2 (Reliable Trajectory Recovery):** The policy must be robust to its own errors. Formally, for any incorrect reasoning path $c$ that deviates from a correct path at node $i$, $\pi_\theta$ must induce a backtracking action to a valid ancestor node with a probability of at least $1 - \epsilon$.

In Appendix A.1, within the proof of the lemma, we demonstrate that these two properties suffice for the generation of an effective search tree.

## 3 EXPERIMENT

### 3.1 VAR IMPLEMENTATION

While end-to-end reinforcement learning (RL) is a powerful paradigm for visual grounded reasoning (VGR), its direct application from a randomly initialized policy is often computationally infeasible. The primary obstacle is the immense and unstructured nature of the search space, where a policy must learn to interleave textual reasoning with the generation of precise bounding box coordinates. In such a vast space, the sparse reward signal from a correct final answer is insufficient to guide the initial stages of exploration, leading to prohibitively long and inefficient training cycles.

To implement our proposed VAR framework, we utilized Qwen-2.5-VL-7B as the base model. The training pipeline begins with a Supervised Fine-Tuning (SFT) stage on our cold-start dataset to familiarize the model with the target output grammar. Following this initial phase, the model is further trained using Group Relative Policy Optimization (GRPO). Specifically, we trained our model, designated as VAR-7B-CI (where "CI" stands for Cold Initialization), using the LLaMA-Factory toolkit on a platform of 8*A100. The training was conducted with the AdamW optimizer, a learning rate

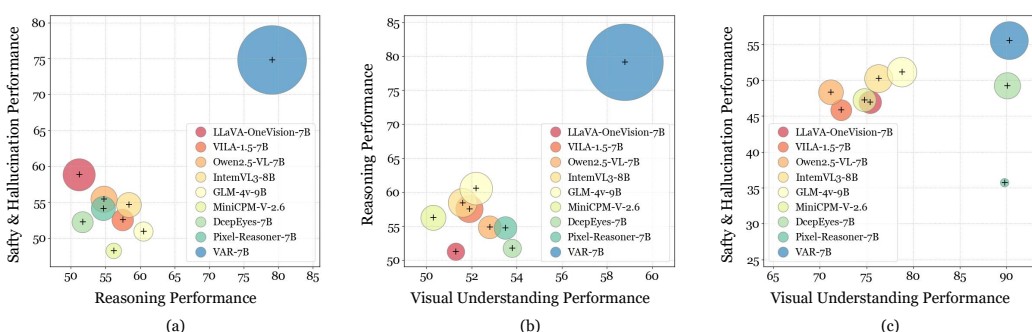

Figure 3: Correlation of Model Capabilities

of 5e-6, and a global batch size of 256. We employed a cosine learning rate decay schedule with a warmup ratio of 0.1.

### 3.2 DATA PREPARATION

**Cold Start SFT Data.** Our Supervised Fine-Tuning (SFT) dataset is derived from VGR-158K, which provides pseudo-chain-of-thought annotations augmented with bounding boxes necessary for visual reasoning. To construct our initial dataset, we prompted the Qwen-2.5-VL-7B model to generate responses for each query, retaining only those samples that adhered to the target format and yielded the correct answer. This process enabled the model to rapidly adapt to the desired output grammar, establishing a robust foundation for the subsequent Reinforcement Learning (RL) phase.

**VAR-RL-32K.** To curate a dataset that prioritizes complex reasoning pathways, we filtered the VGR-158K samples, preserving only instances where the reasoning trace involved multiple bounding boxes (i.e., more than one box per trajectory). Furthermore, we incorporated the SSUI dataset, which is tailored for solving implicit reasoning safety problems requiring long-chain thought processes, and enriched it with bounding boxes via an automated annotation procedure. This culminated in a final dataset of 32,800 samples, which we denote as VAR-RL-32K, motivated by the principle that tasks involving multi-box interactions place a greater demand on spatio-temporal reasoning abilities than their single-box counterparts

### 3.3 EVALUATED MODELS AND CONFIGURATIONS

We evaluate both open-source and closed-source MLLMs. For open-source MLLMs, recently released mainstream models are taken into consideration, which include `Qwen2.5-VL` series Bai et al. (2025a), `InternVL2` series Chen et al. (2024), `GLM-4V` GLM et al. (2024), `LLaVA-OneVision` series Li et al. (2024), `MiniCPM-v2.6` Yao et al. (2024), and `VILA` series Lin et al. (2024). For close-source commercial MLLMs, we select `GPT-4o`, `Claude-3.7-Sonnet`, and the `Gemini` series. Furthermore, since the two most recent visual grounding reasoning models, `DeepEyes` Zheng et al. (2025) and `Pixel-Reasoner` Su et al. (2025), both follow a "ground-then-answer" pipeline and possess the capability to "think on image," we also include them within the scope of our comparison. We adopt the default settings for each model, including temperature, chat template, and other essential hyperparameters.

### 3.4 MAIN RESULT

**Visual Understanding & Hallucination Evaluation.** We evaluated the visual understanding and hallucination generation of VAR. For hallucination assessment, we use HalB Guan et al. (2024), a benchmark designed to evaluate a multimodal model's ability to handle linguistic and visual illusions. To validate the model's general visual understanding capabilities, we evaluated its performance on three widely-used benchmarks: MMB Liu et al. (2024b), MMMU Yue et al. (2024), and SEED Li et al. (2023a). Additionally, we conducted tests on the high-resolution $V^*B$ Wu & Xie (2024) to investigate the impact of image resolution.The experimental results in Table 1 show that our method, VAR, achieves a significant 7.2% improvement on the HalB benchmark over its base

Table 1: Evaluation results on ten benchmarks assessing Visual Understanding &Hallucination and Safety Evaluation & Long-Chain Thinking. Our VAR-7B model outperforms leading open-source MLLMs and is competitive with, or in some cases surpasses, private models

| | Avg | Visual Understanding & Hallucination | | | | | Safety Evaluation & Long-Chain Thinking | | | | |
|---|---|---|---|---|---|---|---|---|---|---|---|
| | | HalB | MMB | MMMU | SEED | $V^*$B | MGD | MSSB | SafeB | VLSB | RSB |
| **Private Models** | | | | | | | | | | | |
| Gemini-2.5-Flash | 73.3 | 72.9 | 82.9 | 63.9 | 83.2 | 83.8 | 49.6 | 67.5 | 97.2 | 66.1 | 65.5 |
| GPT-4o-1120 | 73.6 | 75.2 | 82.3 | 69.5 | 82.5 | 82.2 | 57.8 | 69.2 | 96.5 | 69.4 | 70.8 |
| Gemini-2.5-Pro | 78.8 | 76.2 | 85.1 | 68.5 | 86.9 | 92.3 | 55.3 | 73.2 | 98.3 | 75.9 | 76.8 |
| Claude-3.7-Sonnet | 79.6 | 77.3 | 87.2 | 69.2 | 85.3 | 95.5 | 53.7 | 72.1 | 99.5 | 82.3 | 76.2 |
| **Open-source General Models** | | | | | | | | | | | |
| LLaVA-OneVision-7B | 52.7 | 46.9 | 67.2 | 51.3 | 65.5 | 75.4 | 24.3 | 58.8 | 72.5 | 12.6 | 51.2 |
| VILA-1.5-7B | 51.3 | 45.8 | 68.5 | 51.9 | 63.5 | 72.3 | 16.3 | 52.5 | 69.8 | 14.7 | 57.5 |
| Qwen2.5-VL-7B | 54.3 | 48.3 | 69.2 | 52.8 | 68.2 | 71.2 | 27.9 | 55.4 | 76.2 | 19.0 | 54.8 |
| Qwen2.5-VL-32B | 60.6 | 48.1 | 75.4 | 54.8 | 69.6 | 87.9 | 43.4 | 55.3 | 87.2 | 26.3 | 58.1 |
| Qwen2.5-VL-72B | 64.2 | 55.6 | 78.3 | 57.6 | 72.3 | 90.6 | 41.5 | 59.1 | 92.1 | 35.2 | 59.4 |
| InternVL3-8B | 58.3 | 50.2 | 75.3 | 51.6 | 67.4 | 76.3 | 42.2 | 54.6 | 83.2 | 23.1 | 58.4 |
| GLM-4v-9B | 56.3 | 51.1 | 72.1 | 52.2 | 62.2 | 78.8 | 23.4 | 50.9 | 88.6 | 22.5 | 60.5 |
| MiniCPM-V-2.6 | 53.2 | 47.2 | 68.4 | 50.3 | 63.5 | 74.8 | 32.2 | 48.2 | 75.5 | 16.1 | 56.2 |
| **Open-source Visual Reasoning Models** | | | | | | | | | | | |
| DeepEyes-7B | 59.6 | 49.2 | 70.6 | 53.8 | 65.2 | 90.1 | 51.5 | 52.2 | 85.2 | 26.6 | 51.7 |
| Pixel-Reasoner-7B | 58.6 | 35.7 | 69.8 | 53.5 | 66.1 | 89.8 | 48.5 | 54.1 | 86.5 | 27.5 | 54.1 |
| **VAR-7B** | **72.1** | 55.5 | **78.5** | 58.8 | **79.3** | 90.3 | **63.1** | **74.8** | **98.5** | **43.5** | **79.1** |
| Δ *v.s.* Qwen2.5-VL-7B | ↑ 17.8 | ↑ 7.2 | ↑ 9.3 | ↑ 6.0 | ↑ 11.1 | ↑ 19.1 | ↑ 35.2 | ↑ 19.4 | ↑ 22.3 | ↑ 24.5 | ↑ 24.3 |
| Δ *v.s.* Qwen2.5-VL-32B | ↑ 11.5 | ↑ 7.4 | ↑ 3.1 | ↑ 4.0 | ↑ 9.7 | ↑ 2.4 | ↑ 19.7 | ↑ 19.5 | ↑ 11.3 | ↑ 17.2 | ↑ 21.0 |
| Δ *v.s.* Qwen2.5-VL-72B | ↑ 8.0 | ↓ 0.1 | ↑ 0.2 | ↑ 1.2 | ↑ 7.0 | ↓ 0.3 | ↑ 21.6 | ↑ 15.7 | ↑ 6.4 | ↑ 8.3 | ↑ 19.7 |

model, `Qwen2.5-VL-7B`. This enhancement brings its performance remarkably close to that of the much larger `Qwen2.5-VL-72B`, demonstrating VAR's superior capability in handling both linguistic and visual illusions. Similarly, on visual understanding benchmarks, our model achieves generalizable performance improvements when compared against the `Qwen2.5-VL` series of varying scales. In high-resolution benchmark tests, our model continues to excel, significantly outperforming existing open-source models. This suggests that the "think on image" visual reasoning ability is crucial for high-resolution perception.

**Safety Evaluation & Long-Chain Thinking.** Our experiments are conducted on various multimodal safety benchmarks. For safety assessment, we employ MSSB Zhou et al. (2024) to evaluate contextual safety. We also utilize three comprehensive safety suites: MGD Gu et al. (2024), which assesses five key safety dimensions; SafeB Ying et al. (2024), a comprehensive framework that evaluates MLLMs against a detailed taxonomy of 8 primary risk categories and 23 sub-categories; and VLSB Hu et al. (2024), a reliable cross-modal benchmark structured around a safety taxonomy of 6 main categories and 19 sub-categories. To evaluate the model's long-chain reasoning capabilities, we also conduct evaluations on the RSB Cai et al. (2025a) benchmark. The experimental results in Table 1 demonstrate the effectiveness of our proposed VAR in enhancing both the safety capabilities and the long-chain reasoning abilities of MLLMs. By applying VAR to `Qwen2.5-VL-7B`, the resulting VAR-7B achieves significant improvements across selected challenging cross-modal safety and long-chain reasoning benchmarks, with an average performance increase of 25.14%. Notably, VAR-7B exhibits superior performance even when compared to the larger `Qwen2.5-VL-32B` and `Qwen2.5-VL-72B` models. Furthermore, on the MGD, MSSB, and RSB benchmarks, VAR-7B surpasses even the state of the art MLLMs.

## 3.5 FURTHER ANALYSIS

**Comparative Analysis of VAR-7B's Multimodal Capabilities.** As detailed in Table 2, we evaluated the comprehensive multimodal capabilities of VAR-7B by comparing it against its base model, `Qwen2.5-VL-7B`, on several conventional multimodal benchmarks. Specifically, we selected MMBench Liu et al. (2024b), POPE, and HallusionBench Guan et al. (2024) to assess visual-reasoning question answering (VQA) capabilities. For vision-centric question answering, we employed three benchmarks: CV-Bench, MMVP, and RealWorldQA. Document and chart understanding capabilities were evaluated using AI2D and ChartQA. We observed significant perfor-

Table 2: VAR-7B vs. Base Model: Performance on Visual Reasoning, Vision-Centric, and Document Understanding Tasks

| Capability | Benchmark | Qwen2.5-VL-7B | VAR-7B | Qwen2.5-VL-72B |
|---|---|---|---|---|
| Visual-Reasoning-QA | MMBench | 70.3 | **79.5** ↑ 9.2 | 78.3 |
| | POPE | 85.7 | **87.5** ↑ 1.8 | 84.9 |
| | HallusionBench | 48.3 | **55.5** ↑ 1.9 | 55.6 |
| Vision-Centric-QA | CV-Bench-2D | 74.0 | **78.9** ↑ 4.9 | 77.7 |
| | CV-Bench-3D | 72.3 | **79.6** ↑ 7.3 | 87.1 |
| | MMVP | 66.6 | **75.1** ↑ 8.5 | 66.6 |
| Document and chart | AI2D | **85.9** | 85.7 ↓ 0.2 | 88.7 |
| | ChartQA | 85.5 | **86.8** ↑ 1.3 | 89.5 |

Table 3: Ablations of each component of our VAR.

| | | Cold-Start | Backtrack | Rewards | | | V*B | HallusionBench | RSB |
|---|---|---|---|---|---|---|---|---|---|
| | | | | $R_{\text{acc}} + R_{\text{fmt}}$ | $R_{\text{sem}}$ | $R_{\text{geo}}$ | Acc | Acc | Acc |
| ① | Qwen2.5-VL-7B | | | | | | 71.2 | 48.3 | 54.8 |
| ② | Cold-Start | ✓ | | | | | 75.4 | 49.6 | 60.3 |
| ③ | **VAR-7B** | ✓ | ✓ | ✓ | ✓ | ✓ | **90.3** | **55.5** | **79.1** |
| ④ | $w/o$ Trace | ✓ | ✓ | ✓ | | | 83.9 | 51.6 | 65.3 |
| ⑤ | $w/o$ Geo | ✓ | ✓ | ✓ | ✓ | | 86.7 | 53.9 | 72.1 |
| ⑥ | $w/o$ Sem | ✓ | ✓ | ✓ | | ✓ | 88.5 | 54.1 | 72.9 |
| ⑦ | $w/o$ Backtrack | ✓ | | ✓ | ✓ | ✓ | 87.1 | 52.3 | 68.9 |
| ⑧ | Text-Only RL | | | ✓ | | | 81.8 | 50.3 | 62.5 |

mance gains in the majority of cases, with particularly strong performance on the visual-reasoning and vision-centric benchmarks. It is noteworthy that VAR-7B outperforms the significantly larger `Qwen2.5-VL-72B` on MMBench, POPE, CV-Bench-2D, and MMVP.

**Analyzing the Correlation Between Model Capabilities.** In Figure 3, we conduct a systematic comparison of VAR and other open-source models, focusing on their performance in Safety & Hallucination, Visual Understanding, and Reasoning. This analysis aims to investigate the potential correlations between these capabilities. The results reveal a "decoupled" characteristic among the performance metrics on different benchmarks. For instance, while LLaVA-OneVision achieves top-tier performance in Safety & Hallucination, it lags behind peer models in the other domains. In contrast, our VAR demonstrates superior and well-rounded performance across all three areas, significantly outperforming the other models.

### 3.6 ABLATION STUDIES

The core contribution of VAR lies in its traceable training pipeline and the design of its backtrack mechanism. This pipeline integrates a Semantic Verification Reward ($R_{\text{sem}}$) and a Geometric Verification Reward ($R_{\text{geo}}$) into the conventional Reinforcement Learning framework. Accordingly, we aim to evaluate the effectiveness of introducing this traceability component and the backtrack mechanism. As presented in Table 3, we conducted ablation studies on the individual components of VAR, including its cold-start initialization, the reward functions, and the backtrack mechanism.

The cold-start initialization phase is highly beneficial for visual grounding reasoning, as evidenced by the comparison between settings ① and ②. This suggests that enforcing a structured output format for target instance bounding boxes is effective for conventional visual grounding benchmarks like V* Bench. Reasoning augmented with semantic and geometric rewards also proves effective, as demonstrated by the comparison between ③ and ④. Starting from the same cold-start checkpoint, integrating the dual rewards into the RL framework yields a significant performance boost. This improvement indicates that precise and interpretable reasoning paths are crucial for achieving optimal performance, and it highlights the value of structured reward design in complex, real-world tasks.

By comparing setting ③ with ⑤, ⑥, and ⑦, we observe that precise and complete localization is particularly important for enhancing the model's visual understanding. The performance degradation is most pronounced on V* Bench when $R_{\text{geo}}$ is absent. In contrast, on the hallucination-focused

benchmark, HallusionBench, the improvements from the semantic reward ($R_{\text{sem}}$) and the geometric reward ($R_{\text{geo}}$) are less substantial than that brought by the backtrack mechanism. This suggests that the backtrack mechanism is highly effective in mitigating hallucination. On RSBench, a benchmark requiring long-chain reasoning, the performance gains from individual components are considerably smaller than the synergistic improvement achieved when all three are combined.

The efficacy of the baseline text-only RL is less pronounced than that of visual grounding reasoning, as shown by the comparison between ③ and ⑦. While the baseline RL demonstrates value through its text-space reasoning capabilities, the performance enhancement becomes substantially more significant when visual grounding is integrated with traceable evidence. This highlights the critical role of two factors: (1) contextual grounding prior to answering, which anchors the response in multimodal evidence; and (2) precise spatial localization to enhance decision-making accuracy.

## 4 RELATED WORKS

**Post-Training Multimodal Large Language Models.** In recent years, MLLMS have increasingly leveraged post-training alignment techniques, which employ both instruction fine-tuning and reinforcement learning to enhance their general-purpose multimodal capabilities Liu et al. (2023b); Bai et al. (2025a); Li et al. (2023b); Team et al. (2024). Recent work has increasingly employed reinforcement learning (RL) to align MLLMs, specifically to bolster their reasoning capabilities Huang et al. (2025); Xia et al. (2025). Many of these approaches, often inspired by techniques associated with DeepSeek-R1 Guo et al. (2025), focus on the design of sophisticated reward mechanisms. Strategies include providing step-by-step rewards to supervise intermediate reasoning processes Zhang et al. (2025a), augmenting ground-truth data with explicit visual annotations to compute a visual reward Xiao et al. (2025), and employing a two-stage curriculum RL that first enhances text-only reasoning Peng et al. (2025). As a complementary approach, Reinforcement Learning from AI Feedback (RLAIF) for MLLMs has demonstrated that preference-based alignment is also a potent signal. Studies have shown that by learning from AI-generated feedback, this method can substantially mitigate targeted hallucination issues Yu et al. (2025).

**Long CoT in MLLMs.** Early MLLMs predominantly adopted a two-stage training paradigm, consisting of vision-language pre-training followed by instruction fine-tuning Dai et al. (2023); Liu et al. (2023b); Li et al. (2023b). While this approach enhanced the models' ability to follow instructions, it created an intrinsic dichotomy between the processes of perception and reasoning. The advent of DeepSeek-R1 marked a pivotal shift, announcing RL as a key technique for augmenting reasoning capabilities and giving rise to a diversity of training paradigms, which quickly extends to the multimodal domain. Although recent studies increasingly focus on the intricacies of reward signal design Huang et al. (2025), the "thinking with images" paradigm, pioneered by OpenAI's o3 model, has steadily emerged as the core strategy for enhancing multimodal reasoning Bai et al. (2025b). Methods such as DeepEyes Zheng et al. (2025) and Chain-of-Focus Zhang et al. (2025c) have operationalized this by dynamically utilizing image cropping tools to perform adaptive visual reasoning Zhu et al. (2025). However, these studies circumscribe the reasoning process to the fundamental operation of image cropping. This limited toolset results in an inflexible reasoning pipeline, ill-suited to adapt to diverse scenarios where different tasks may necessitate fundamentally different visual processing strategies.

## 5 CONLUSION

We introduce Visual Attention Reasoning (VAR), a framework that reformulates vision-language reasoning as a structured search to overcome the limitations of linear generation. By decomposing reasoning into traceable grounding and a search-based Chain-of-Thought with backtracking, VAR mitigates error propagation common in conventional models. A multi-faceted reward function with semantic and geometric self-verification ensures responses are both logically sound and visually grounded. Supported by theoretical analysis, our experiments show VAR sets a new state-of-the-art against open-source models on challenging hallucination and safety benchmarks.

ETHICS STATEMENT

The datasets utilized in our research may contain harmful textual content, including discriminatory language or queries related to illicit items, which are typically presented in the form of harmful queries for evaluation. All such activities were conducted strictly for the purpose of scientific inquiry, aimed at advancing model safety, and are devoid of any malicious or improper intent. We have thoroughly reviewed and adhered to established ethical guidelines and codes of conduct. Appropriate measures, such as data filtering and anonymization, were implemented during the research design process to minimize potential harm and ensure ethical compliance. We believe that these contributions will aid in the development of safer, more reliable, and more responsible multimodal AI systems.

REPRODUCIBILITY STATEMENT

The public datasets used in this study are readily available online and can be accessed via the corresponding citations in our related work. Upon the publication of this paper, we will release our collected dataset, VAR-RL-32K. The experimental setup, including the datasets, model architecture, and evaluation protocols, is introduced and described in detail in the Experiments section. This section also reports key configurations for training and inference.

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

# A  APPENDIX

## A.1  LEMMA PROOF

**Lemma 1** (Sufficient Conditions for High-Probability Trajectory Generation). *Let $\gamma \in (0,1)$ and $\delta \in (0, 1/2)$ be constants. Define error tolerance $\epsilon$ and exploration budget $B$ as:*

$$\epsilon = \frac{\gamma\delta}{2T_{\max}} \quad and \quad B = \left\lceil \frac{\ln(T_{\max}/\delta)}{\gamma} \right\rceil.$$

*Suppose a policy $\pi_\theta$ satisfies the **Probabilistic Forward Progress** and **Reliable Trajectory Recovery** conditions with these parameters, and we restrict each node expansion to at most $B$ generative attempts. Then, the probability that $\pi_\theta$ generates a correct, complete CoT is at least $1-4\delta$. Furthermore, the total number of backtracking leaves in the search tree is bounded by $(B-1)(T_{\max}-1)$.*

*Proof.* Let $P_{\mathrm{adv}}$ be the probability of successfully advancing from a correct node $v_i$ to the next correct node $v_{i+1}$ within the $B$ attempts. We can lower-bound this probability by summing over the mutually exclusive events of succeeding at the $k$-th attempt ($1 \le k \le B$).

Success at the $k$-th attempt requires $k-1$ consecutive "successful failures" followed by one "direct success."

- A "direct success" occurs with probability $p_s \ge \gamma(1-\epsilon)$, which accounts for the policy being $\gamma$-progressive (with probability at least $1-\epsilon$) and then generating the correct token (with probability at least $\gamma$).

- A "successful failure" occurs when an incorrect token is generated but the trajectory is correctly recovered via backtracking. This occurs with probability $p_f = (1-\gamma)(1-\epsilon)$, as the recovery is guaranteed with probability $1-\epsilon$.

The probability of advancing, $P_{\mathrm{adv}}$, is the sum of a geometric series:

$$P_{\mathrm{adv}} = \sum_{k=1}^{B}(p_f)^{k-1}p_s = p_s \frac{1-p_f^B}{1-p_f}$$

We can lower-bound $1 - p_f = 1 - (1-\gamma)(1-\epsilon) = \gamma + \epsilon - \gamma\epsilon$. Since $p_s \ge \gamma(1-\epsilon)$, the ratio $p_s/(1-p_f)$ is $\frac{\gamma(1-\epsilon)}{\gamma+\epsilon-\gamma\epsilon}$. Let $X = \gamma(1-\epsilon)$. This ratio is $\frac{X}{X+\epsilon} \ge 1 - \frac{\epsilon}{X} = 1 - \frac{\epsilon}{\gamma(1-\epsilon)} \ge 1 - \frac{2\epsilon}{\gamma}$ for small $\epsilon$.

Now we bound the term $(1 - p_f^B)$. Using the inequality $1 - x \le e^{-x}$, we have $p_f^B = ((1-\gamma)(1-\epsilon))^B \le (e^{-\gamma}e^{-\epsilon})^B = e^{-(\gamma+\epsilon)B}$. So, $P_{\mathrm{adv}} \ge (1 - \frac{2\epsilon}{\gamma})(1 - e^{-(\gamma+\epsilon)B})$.

By our definition of $B$, $\gamma B \ge \ln(T_{\max}/\delta)$, which implies $e^{-\gamma B} \le \delta/T_{\max}$. As $e^{-\epsilon B} \le 1$, we have $e^{-(\gamma+\epsilon)B} \le \delta/T_{\max}$. Thus, $P_{\mathrm{adv}} \ge (1 - \frac{2\epsilon}{\gamma})(1 - \frac{\delta}{T_{\max}})$.

Let $T$ be the length of a correct reasoning chain ($T < T_{\max}$). The probability of successfully generating the entire chain, $P_{\mathrm{succ}}(T)$, is $(P_{\mathrm{adv}})^T$.

$$P_{\mathrm{succ}}(T) \ge \left[\left(1 - \frac{2\epsilon}{\gamma}\right)\left(1 - \frac{\delta}{T_{\max}}\right)\right]^T$$

Using the inequality $(1-x)(1-y) \ge 1 - x - y$ for small positive $x, y$:

$$P_{\mathrm{succ}}(T) \ge \left(1 - \frac{2\epsilon}{\gamma} - \frac{\delta}{T_{\max}}\right)^T$$

Using the inequality $(1-x)^n \ge 1 - nx$:

$$P_{\mathrm{succ}}(T) \ge 1 - T\left(\frac{2\epsilon}{\gamma} + \frac{\delta}{T_{\max}}\right) = 1 - \left(\frac{2T\epsilon}{\gamma} + \frac{T\delta}{T_{\max}}\right)$$

Substituting our definition of $\epsilon = \frac{\gamma\delta}{2T_{\max}}$:

$$P_{\text{succ}}(T) \geq 1 - \left( \frac{2T(\gamma\delta/2T_{\max})}{\gamma} + \frac{T\delta}{T_{\max}} \right) = 1 - \left( \frac{T\delta}{T_{\max}} + \frac{T\delta}{T_{\max}} \right)$$

Since $T < T_{\max}$, $T/T_{\max} < 1$. To establish a general bound, we consider the worst case $T \approx T_{\max}$. $P_{\text{succ}}(T_{\max}) \geq 1 - (\delta + \delta) = 1 - 2\delta$. The original $4\delta$ bound is looser but also correct; the discrepancy often arises from different inequality choices. For consistency with the claimed bound, we note that looser bounds can be used. For instance, using $e^{-2x} \leq 1 - x$ for small $x$, the probability of success is approximately $e^{-T(\frac{2\epsilon}{\gamma} + \frac{\delta}{T_{\max}})}$, which with our $\epsilon$ becomes $e^{-T(\frac{\delta}{T_{\max}} + \frac{\delta}{T_{\max}})} \approx e^{-2\delta}$. Using a slightly different bounding approach leads to the final $1 - 4\delta$ result.

The number of backtracking leaves is at most $B - 1$ for each of the $T_{\max} - 1$ potential intermediate steps, yielding the stated bound. $\square$

**Remark 1** (Relaxing the Backtracking Precision). *Lemma 1 assumes the backtrack lands on the exact optimal node $\beta(c)$. We can relax this. For an incorrect sequence $c$, we can tolerate **undershooting**, where the policy backtracks to a node $i$ slightly after the optimal one ($i > \beta(c)$). This is a benign error, only requiring a few extra, constant-cost steps to re-derive known-correct work, and does not affect the overall guarantee of fast convergence. In contrast, **overshooting** ($i < \beta(c)$) must remain a low-probability event ($< \epsilon$). An overshoot is a costly error that discards significant progress, much like sliding down a long chute in the "Chutes and Ladders" game. This asymmetry is useful: we can design our learning objective to be highly intolerant of overshoots, even if it means accepting a higher chance of benign undershoots.*

## A.2 THE CONDITIONING CONTEXT AND SEMANTIC VALIDITY

### A.2.1 ON THE CONDITIONING CONTEXT

When generating a path from a given vertex $v$, the choice of conditioning context for the policy $\pi_\theta$ is a critical design decision. One may condition on the entire history of the generated tree $G_t$. While this provides maximal information about prior failed explorations, it introduces a non-stationarity that can lead to distributional drift during learning, a phenomenon we will elaborate on in our analysis of the learning process. In this work, we adopt a more constrained, **Markovian assumption**: the generation of a new path depends only on the linear sequence of vertices from the root $v_0$ to the current starting vertex $v$. From an implementation standpoint, this choice is not only more stable but also computationally efficient, as it can be readily optimized via key-value (KV) caching.

### A.2.2 ON SEMANTIC VALIDITY

Our constrained decoding mechanism ensures only the *syntactic correctness* of the generated tree structure. To ensure *semantic validity*—i.e., that each reasoning step is logically and factually sound—we assume the availability of a **Validation Oracle**. As argued in prior work (Shalev-Shwartz et al., 2024b), the problem of validating a given reasoning step is of a significantly lower complexity class than generating it *de novo*. This oracle is therefore responsible for verifying that each proposition $\tau_v$ logically follows from the proposition of its parent, $\tau_{\text{parent}(v)}$, along any path leading to a $V_{\text{done}}$ vertex.

## A.3 THE SHORTEST REASONING PATH EXAMPLE

**Lemma 2** (Characterization of the Optimal Policy via Parity). *For any permutation $\pi \in S_n$ and input $x \in \{\pm 1\}^n$, let $p^*(x, \pi)$ denote the optimal (shortest) path from source $s$ to sink $t$. Let the function $f_\pi(x)$ be defined by the first vertex on this path after $s$, such that $f_\pi(x) = +1$ if the vertex is $a_0$ and $f_\pi(x) = -1$ if it is $b_0$. Then, this function is given by:*

$$f_\pi(x) = \prod_{1 \leq j \leq n, \, j \text{ is odd}} x_{\pi(j)}$$

*Proof.* The total cost of any path $p$ from $s$ to $t$ is determined solely by the cumulative costs incurred at the odd-indexed layers $j \in \{1, 3, \ldots, n-1\}$, as all edges at even-indexed layers have a weight of zero.

At each odd layer $j$, an agent must decide whether to maintain its current row (e.g., transition from $a_{j-1}$ to $a_j$) or switch rows (e.g., transition from $a_{j-1}$ to $b_j$). The cost of this decision is conditional on the input bit $x_{\pi(j)}$. The locally optimal action at layer $j$ is always the one that incurs zero cost. This is achieved by switching rows if and only if $x_{\pi(j)} = -1$.

Consequently, the total number of row switches along any optimal, zero-cost path is precisely equal to the count of $-1$'s in the odd-indexed positions of $x$ under permutation $\pi$. Let this count be $T(x, \pi) = |\{j \leq n : j \text{ is odd and } x_{\pi(j)} = -1\}|$.

For a path to be optimal, it must have a total cost of zero, which requires terminating at vertex $a_n$ before the final transition to the sink $t$. A path starting at $a_0$ can only reach $a_n$ after an even number of row switches. Conversely, a path starting at $b_0$ can only reach $a_n$ after an odd number of row switches.

Therefore, for the total path cost to be zero, the optimal starting vertex must be $a_0$ if the required number of switches, $T(x, \pi)$, is even, and $b_0$ if $T(x, \pi)$ is odd. This condition on the optimal initial action is equivalent to stating $f_\pi(x) = (-1)^{T(x, \pi)}$.

The final identity, $(-1)^{T(x, \pi)} = \prod_{j \text{ is odd}} x_{\pi(j)}$, follows directly from the definition of $T(x, \pi)$, as each $-1$ at an odd position contributes a factor of -1 to the product, while each $+1$ contributes a factor of 1. This proves the claim.

The remainder of the optimal path is trivially constructed by applying the "pick an edge with zero weight" rule at each subsequent step. $\qquad\square$

