# OpenReview forum: "VAR: Visual Attention Reasoning via Structured Search and Backtracking"
_ICLR.cc/2026/Conference — ICLR 2026 Conference Withdrawn Submission_

### Official Review · Reviewer_QgDZ · 2025-10-30

**Soundness:** 3
**Presentation:** 3
**Contribution:** 2
**Rating:** 6
**Confidence:** 4

**Summary:**

The paper proposes Visual Attention Reasoning (VAR), a post-training recipe that involves visual grounding into chain-of-thought verification. The experimental results across diverse benchmarks support the effectiveness of their approach.

**Strengths:**

- Benchmarking. They do sufficient experiments on multiple datasets and demonstrate consistent performance gains against baseline (Qwen2.5-VL-7B) on more than 10 datasets.
- Presentation. The presentation of this paper is clear.

**Weaknesses:**

- Weak Novelty. The design from this paper involves **backtracking**, while this core idea is not new. Authors avoid discussing the difference of VAR against those papers, either from the perspective of hallucination [A] or reasoning [B]. Authors are suggested to be more clear on such aspects.

- Comparisons are not that convincing. Note that authors mainly compare two types of models in their main tables, one is instruct models such as InternVL3 or Qwen2.5-VL, the other is Visual Reasoning models including DeepEyes and PixelReasoner.

   + For instruct models, it should be pointed out that direct comparisons to non post-trained models are not fair. Authors are suggested to compare VAR with models that are post-trained by GRPO, such as VL-Rethinker [B], or Revisual-R1 [C]. How would VAR compare to these models on the used benchmarks in this paper?

   + For visual reasoning models like DeepEyes and PixelReasoner, it should be highlighted that they excel at visual searching in high-resolution images. I notice that authors provide the performance gains on V* Bench, which satisfies this application, but the performance gain is only 0.2%. How is the performance of VAR-7B on HR-Bench [D] against DeepEyes, for example?

[A] Look Twice Before You Answer: Memory-Space Visual Retracing for Hallucination Mitigation in Multimodal Large Language Models. ICML.
[B] VL-Rethinker: Incentivizing Self-Reflection of Vision-Language Models with Reinforcement Learning. NeurIPS 2025.
[C] Advancing Multimodal Reasoning: From Optimized Cold Start to Staged Reinforcement Learning.
[D] Divide, Conquer and Combine: A Training-Free Framework for High-Resolution Image Perception in Multimodal Large Language Models.

**Questions:**

Please see my weakness 1 and 2. I will consider raise my score only if authors clearly address the concerns.

---

### Official Review · Reviewer_c2sS · 2025-11-01

**Soundness:** 2
**Presentation:** 3
**Contribution:** 2
**Rating:** 4
**Confidence:** 3

**Summary:**

This paper proposes a hallucination mitigation approach based on structured search and backtracking, using multi-component rewards to instill a structured reasoning paradigm. The experimental results indicate strong improvements in visual–semantic understanding for MLLMs. However, the paper’s use of problem search and reinforcement learning is already prevalent in prior work. The manuscript does not sufficiently explain why it achieves superior results, and the comparative experiments appear insufficient in coverage and depth.

**Strengths:**

1. The proposed multi-reward mechanism broadly covers potential failure modes that can lead to hallucinations.
2. The paper uses clear and well-structured figures and text to precisely present the motivation, proposed solution, experimental setup, and procedure.
3. The proposed Visual Attention Reasoning (VAR) workflow is relatively novel and aligns with current mainstream approaches to mitigating hallucinations.
4. The experimental results are strong, demonstrating the effectiveness of the method.

**Weaknesses:**

1. Limited novelty at the module level: Many submodules resemble prior work in large-model training and fine-tuning; the contribution lies more in the combination and workflow design than in entirely new primitives.
2. Insufficient baselines: The comparative setup is relatively simple and lacks systematic comparisons against state-of-the-art hallucination-mitigation methods.
3. High computational and latency cost: The four-fold reward computation and the search/backtracking mechanism are time-consuming. Although some accelerations are provided, both training and inference remain costly.
4. Insufficient sensitivity analysis: The paper does not thoroughly study sensitivity to reward weights (e.g., α, β, γ) and search budgets (e.g., path length, number of backtracks), leaving questions about stability and reproducibility.

**Questions:**

1. Please clearly describe the main differences between your method and existing approaches such as consistency verification, iterative checking, multi-agent workflows, and confidence-based reinforcement learning, and explain the specific advantages of your proposed approach.
2. More comprehensive comparisons with relevant and recent methods are needed to demonstrate that your approach indeed achieves better performance in mitigating hallucination problems.

---

### Official Review · Reviewer_c8AX · 2025-11-01

**Soundness:** 2
**Presentation:** 2
**Contribution:** 3
**Rating:** 2
**Confidence:** 3

**Summary:**

This paper proposes VAR, a CoT inference paradigm for MLLMs that generates detailed textual descriptions of images integrated with object bounding box coordinates. Guided by newly introduced semantic verification and geometric verification rewards, this design grounds the model's reasoning processes in visual information. Additionally, VAR employs a depth first search tree to maintain reasoning trajectories, enabling self-correction and backtracking to previous nodes. This allows the model to search for an optimal reasoning path. The model is trained using a two-stage process: first, finetuning on a newly constructed visual grounding reasoning dataset, followed by training with GRPO. Experimental evaluations on visual understanding, hallucination, safety, and long-chain reasoning benchmarks demonstrate the effectiveness of VAR compared to baseline models and other SOTA methods.

**Strengths:**

Overall, the paper is well-written with a clear logical flow. It proposes semantic verification and geometric verification rewards to enhance visual grounded reasoning, offering valuable insights for the research community. The authors also filtered and constructed a new SFT dataset for visual reasoning. Open sourcing the dataset and data construction pipeline would also be beneficial to other researchers in the field.

**Weaknesses:**

The paper presents a depth first search tree to maintain reasoning trajectories, allowing the model to take multiple reasoning paths and select the most ideal solution. However, reasons for choosing a tree-like structure can be further explored. Alternative reasoning trajectory designs such as graphs could be contrasted with the tree structure empirically to evaluate the advantages of the design of choice.

**Questions:**

1. Some Qwen2.5 VL baseline accuracies presented in table 1 are remarkably lower compared to the results from original Qwen2.5 VL paper.  For example, in table 1, the MMMU benchmark accuracy is 52.8 and 57.6 for Qwen-2.5-VL-7B and Qwen-2.5-VL-72B respectively. However, the results are reported as 58.6 and 70.2 in the original Qwen-2.5-VL paper.
2. Could the authors also provide the SOTA performance results on evaluation benchmarks presented in table 2?
3. The VAR method is only trained and evaluated using Qwen 2.5 VL base model. It would strengthen the validity of VAR design if other base models such as InternVL 3 can also be trained and evaluated.
4. The tree structure as depicted in figure 2 appears to be highly similar as compared to the paper "Multimodal Chain-of-Thought Reasoning: A Comprehensive Survey" figure 3. However, no credit is given to the original paper. Could the authors describe how the drawing is derived in figure 2?
5. The authors provided theoretical proof of the upper bound for the number of backtracing during model reasoning. Could the authors also provide an empirical result of the model inference speed during evaluation? How fast is VAR compared to the baseline model and other SOTA?

**Details Of Ethics Concerns:**

The tree structure in Figure 2 closely resembles Figure 3 from the paper 'Multimodal Chain-of-Thought Reasoning: A Comprehensive Survey' However, the original source is not cited.

---

### Note · Authors · 2025-11-21

I have read and agree with the venue's withdrawal policy on behalf of myself and my co-authors.